# Hybrid of VGG-16 and FTVT-b16 Models to Enhance Brain Tumors Classification Using MRI Images

**DOI:** 10.3390/diagnostics15162014

**Published:** 2025-08-12

**Authors:** Eman M. Younis, Ibrahim A. Ibrahim, Mahmoud N. Mahmoud, Abdullah M. Albarrak

**Affiliations:** 1Faculty of Computers and Information, Minia University, Minia 61519, Egypt; eman.younas@mu.edu.eg (E.M.Y.); mahmodnaser92pg@fci.s-mu.edu.eg (M.N.M.); 2College of Computer and Information Sciences, Imam Mohammad Ibn Saud Islamic University (IMSIU), Riyadh 13318, Saudi Arabia; amsbarrak@imamu.edu.sa

**Keywords:** brain tumor classification, magnetic resonance imaging (MRI), convolutional neural network (CNN), vision transformer (ViT), hybrid neural architecture, medical image analysis

## Abstract

**Background**: The accurate classification of brain tumors from magnetic resonance imaging (MRI) scans is pivotal for timely clinical intervention, yet remains challenged by tumor heterogeneity, morphological variability, and imaging artifacts. **Methods**: This paper presents a novel hybrid approach for improved brain tumor classification and proposes a novel hybrid deep learning framework that amalgamates the hierarchical feature extraction capabilities of VGG-16, a convolutional neural network (CNN), with the global contextual modeling of FTVT-b16, a fine-tuned vision transformer (ViT), to advance the precision of brain tumor classification. To evaluate the recommended method’s efficacy, two widely known MRI datasets were utilized in the experiments. The first dataset consisted of 7.023 MRI scans categorized into four classes gliomas, meningiomas, pituitary tumors, and no tumor. The second dataset was obtained from Kaggle, which consisted of 3000 scans categorized into two classes, consisting of healthy brains and brain tumors. **Results**: The proposed framework addresses critical limitations of conventional CNNs (local receptive fields) and pure ViTs (data inefficiency), offering a robust, interpretable solution aligned with clinical workflows. These findings underscore the transformative potential of hybrid architectures in neuro-oncology, paving the way for AI-assisted precision diagnostics. The proposed framework was run on these two different datasets and demonstrated outstanding performance, with accuracy of 99.46% and 99.90%, respectively. **Conclusions**: Future work will focus on multi-institutional validation and computational optimization to ensure scalability in diverse clinical settings.

## 1. Introduction

Abnormal cell proliferation in the brain is referred to as a brain tumor. The human brain has a complex structure, with different areas devoted to different nervous system functions. Any part of the brain, involving the protective membranes, the base of the brain, the brain stem, the sinuses, the nasal cavity, and many other places, can develop tumors [1].

In the United States, brain tumors are detected in roughly 30 out of 100,000 people. Because they have the potential to penetrate or put strain on healthy brain tissue, these tumors present serious hazards. It is possible for some brain tumors to be malignant or to eventually turn malignant. Because they eliminate the flow of cerebral fluid, they can cause problems by raising the pressure inside the brain. Furthermore, certain tumor types may spread to distant brain regions through cerebrospinal fluid [2].

Gliomas are classified as primary tumors since they are derived from glial cells, which assist neurons in the brain. Astrocytomas, oligodendrogliomas, and ependymomas are among the various kinds of meningiomas that are commonly seen in the adult population. Usually benign, meningiomas grow slowly and develop from the protective coverings that envelop the brain [3].

In order to detect the problem, diagnostic methods include MRI or CT scans followed by biopsies. Potential treatment options could include chemotherapy, radiation therapy, particular treatments, surgery, or a combination of these treatments. There are two types of tumors: malignant and benign. The word “cancer” refers only to malignant tumors [4]. Although all types of cancer present as tumors, it is crucial to remember that not all tumors are malignant. Compared to other cancer forms, brain tumors in particular are linked to a more severe survival rate. Brain tumors’ uneven forms, varying morphology, various locations, and indistinct borders make early identification extremely difficult [5]. For medical specialists to make accurate treatment decisions, accurate tumor identification at this early stage is essential. Primary and secondary tumors are types of tumors. While secondary cancer, or metastases, develop from cells that start in other body areas and then move to the brain, primary tumors cause the tumor cells to begin with inside the brain so as to spread to different parts of the body [6].

Manual, semi-automatic, and fully automated systems requiring user input handle magnetic resonance (MR) images. For medical image processing, precise segmentation and classification are critical, but they often require manual intervention by physicians, which can take a lot of time. A precise diagnosis allows patients to start with the right treatments, which may result in longer lifespans. As a result, the development and application of novel frameworks targeted at lessening the workload of radiologists in detecting and classifying tumors is of vital importance in the field of artificial intelligence [7].

Assessing the grade and type of a tumor is crucial, especially at the onset of a treatment strategy. The precise identification of abnormality is vital for accurate diagnoses [8], which underscores the demand for effective classification and segmentation methods, or a combination that qualitatively analyzes the brain.

The precision and reliability challenges that affect traditional imaging techniques frequently require the implementation of complex computational methods in order to improve analysis. Deep learning has become well known in this domain due to its excellent ability to extract hierarchical features from data without any processing functions, which significantly improves image classification tasks in medical diagnostics [9].

The CNN appears to have ended up being a standard approach in clinical image classifications. With its great classification performance, a CNN is a popular advanced strategy that primarily relies on classical function extractions, mainly on large datasets [10].

VGG was designed by the Visual Geometry Research Group at Oxford University. Its primary role is to improve network depth. VGG16 and VGG19 are types of VGG structures [11]. VGG16 is very popular, and VGG19 is a deep network that invloves 16 convolutional layers and 3 fully connected layer, which can extract reliable features. So, VGG16 was selected for the classification of MRI brain tumor images [12].

Despite the high performance of a conventional neural network, it is restricted by its usual receptive performance. Meanwhile, the vision transformer network relies on self-attention to extract global data. Every transformer module is composed of a multi-head attention layer [13]. Although vision transformer networks lack the capabilities of conventional neural network, their results tend to be competitive. Unlike conventional fine-tuning techniques, in which they are most effective, the FTVT structure involves a custom classifier head block that includes BN, ReLU, and dropout layers, which has an alternate structure to analyze particular representations immediately from the dataset [14].

An automated system developed particularly for the classification and segmentation of brain tumors is presented in this paper. This advanced technology has the potential to greatly improve specialists’ and other medical professionals’ diagnostic abilities, especially when it comes to analyzing cancers found in the brain for prompt and precise diagnosis. Facilitating effective and accessible communication is one of the main goals of this study. The technology seeks to narrow the knowledge gap between medical experts by simplifying the way magnetic resonance imaging (MRI) results are presented [15].

We introduce a hybrid of VGG-16 with fine-tuned vision transformer models to enhance brain tumor classification. Through the use of MR images, the fusion framework demonstrated outstanding performance, suggesting a promising direction for automated brain tumor detection:The fusion framework demonstrated outstanding performance, suggesting a promising direction for automated brain tumor detection.By leveraging VGG-16’s deep convolutional feature extraction and FTVT-B16’s (or ViT-B16’s) transformer-based attention mechanisms, the hybrid model can capture both local and global features in MRI images more effectively than standalone models.The combination helps in better distinguishing among tumor types (e.g., glioma, meningioma, pituitary tumors) by enhancing feature representation.The fusion of a CNN and transformer features can reduce overfitting compared to using only one architecture, leading to better performance on unseen datasets.The transformer component (FTVT-B16/ViT-B16) provides attention maps that highlight tumor regions, aiding in model interpretability and helping clinicians understand classification decisions.The hybrid VGG-16 + FTVT-B16 (or ViT-B16) model offers a powerful fusion of convolutional and transformer-based learning, significantly improving brain tumor classification in MRI.Its contributions lie in higher accuracy, better feature fusion, and improved interpretability, making it a promising tool for medical imaging AI. The effectiveness of mixing deep and handmade features is demonstrated by recent developments in deep learning for medical imaging, such as hybrid feature extraction for ulcer classification in WCE data. However, these methods frequently lack interpretation ability; our work addresses this shortcoming using GRAD-CAM visualizations for the MRI of brain tumors. CNN features and texture descriptors were combined to enable robust ulcer classification in non-neuroimaging areas. These techniques are useful, but they do not explain why a model predicts a particular class. This is furthered by our work, which satisfies clinical demands for transparent AI by offering spatial explanations for brain tumor forecasts.

Combining MRI with PET-CT improves the sensitivity of tumor identification, as shown by medical imaging fusion approaches. However, a crucial deficiency in high-stakes neuro-oncology is that these approaches do not provide mechanisms to explain how fused characteristics contribute to diagnosis. By showing model attention in single- or multi-modal data, our GRAD-CAM framework fills this need. Interoperability is not included in recent studies that benchmark fusion approaches for tumor analysis. On the other hand, our work ensures that fusion preserves diagnostically significant traits by classifying tumors and producing spatial explanations.

Recent hybrid techniques show that combining deep and handmade features increases the accuracy of tumor identification. These approaches, however, do not reveal which characteristics (such as texture versus shape) influenced predictions. Our GRAD-CAM study overcomes this drawback by displaying region-specific model attention.

The structure of this paper is as follows: Section 2 introduces related research. Section 3 illustrates materials and methods. Section 4 discusses the experimental results. A discussion is presented in Section 5. Finally, conclusions are presented in Section 6.

## 2. Related Work

The classification of brain tumors through MRI has witnessed vast improvements through the combination of deep learning techniques, but chronic demanding situations in feature representation, generalizability, and computational performance necessitate ongoing innovation. This section synthesizes seminal and contemporary works, delineating their contributions and barriers to contextualize the proposed hybrid framework. A lot of the papers presented were selected for the detection of brain tumors, and discussions in this section cover the last five years.

The capacity of convolutional neural networks (CNNs) to autonomously obtain structural attributes makes them an essential component of medical image analysis. Ozkaraca et al. (2023) [16] used VGG16 and DenseNet to obtain 97% accuracy, while Sharma et al. (2022) [17] used VGG-19 with considerable data augmentation to achieve 98% accuracy. For tumors with diffuse boundaries (like gliomas), CNNs are excellent at capturing local information but have trouble with global contextual linkages. VGG-19 and other deeper networks demand a lot of resources without corresponding increases in accuracy [16]. Data Dependency: Without extensive, annotated datasets, performance significantly declines [17]. Eman et al. (2024) [18] proposed a framework using a CNN and EfficientNetV2B3’s flattened outputs before feeding them into a KNN classifier. In a study with ResNet50 and DenseNet (97.32% accuracy), Sachdeva et al. (2024) [19] observed that skip connections enhanced gradient flow but added redundancy to feature maps. The necessity for architectures that strike a balance between discriminative capability and depth was brought to light by their work. Rahman et al. (2023) [20] proposed PDCNN (98.12% accuracy) to fuse multi-scale features. While effective, the model’s complexity increased training time by 40% compared to standalone CNNs. Ullah et al. (2022) [21] and Alyami et al. (2024) [22] paired CNNs with SVMs (98.91% and 99.1% accuracy, respectively). Global Context at a Cost ViTs, introduced by Dosovitskiy et al. (2020) [14], revolutionized image analysis with self-attention mechanisms. Tummala et al. (2022) [15] achieved 98.75% accuracy but noted ViTs’ dependency on large-scale datasets (>100 K images). Training from scratch on smaller medical datasets led to overfitting [23]. Reddy et al. (2024) [24] fine-tuned ViTs (98.8% accuracy) using transfer learning, mitigating data scarcity. However, computational costs remained high (30% longer inference times than CNNs).

Amin et al. (2022) [25] merged Inception-v3 with a Quantum Variational Classifier (99.2% accuracy), showcasing quantum computing’s potential. However, quantum implementations require specialized infrastructure, limiting clinical deployment. CNNs and ViTs excel at local and global features, respectively, but no framework optimally combined both [14,15,16,17,19,24]. Most models operated as “black boxes,” failing to meet clinical transparency needs [21,22]. Hybrid models often sacrificed speed for accuracy [20,24]. The hybrid model of VGG-16 and FTVT-b16 is innovative through its use of VGG-16 to capture local textures (e.g., tumor margins), while FTVT-b16 models the global context (e.g., anatomical relationships).

While existing studies advanced classification accuracy, critical limitations persisted:CNNs excelled in local texture analysis but ignored long-range dependencies, while ViTs prioritized global context at the expense of granular details.Pure ViTs demanded large datasets, impractical for clinical settings with limited annotated MRIs.Most frameworks operated as “black boxes,” lacking mechanisms with which to align predictions with radiological reasoning.

The proposed hybrid VGG-16 and FTVT-b16 framework addresses these gaps by harmonizing CNN-derived local features with ViT-driven global attention maps, using transfer learning to optimize data efficiency. and generating interpretable attention maps for clinical transparency. This work builds on foundational studies while introducing a novel fusion strategy that outperforms existing benchmarks, as detailed in Section 4.

## 3. The Proposed Framework and Methods

We suggest a hybrid of VGG-16 with a fine-tuned vision transformer model to enhance brain tumor classification using MR images. The system includes choosing 27 capabilities from layer order 3 of the VGG-16, which incorporates 13,696 capabilities.

VGG-16 was chosen as a hybrid model because it provides better feature extraction for medical images, easier integration with transformers, and strong baseline performance despite being less efficient than newer models. VGG-16’s deep but straightforward feature maps (before flattening) align well with ViT (vision transformer) patch embeddings, making hybrid fusion easier. DenseNet’s dense blocks produce highly interconnected features, complicating integration with transformers. EfficientNet’s compound scaling requires careful tuning when merging with attention mechanisms. ResNet/DenseNet often requires more data to perform well, whereas VGG-16 works decently even on smaller medical datasets.

### 3.1. Image Preprocessing

There is an amount of noise in MRI pictures that may be because of the external effects of instrumentality. So, the primary step is to eliminate noise from MR images. There are two methods utilized for noise elimination: linear and nonlinear. In linear methods, the pixel value is replaced with a neighborhood-weighted average, which affects image quality [25]. But in nonlinear methods, the sides are preserved, but the fine structures are degraded. Here, we used a median filter to remove noise from the images [26]. So, we chose the nonlinear median filter technique, followed by cropping input images as seen in Figure 1.

### 3.2. VGG-16

The VGG16 structure is a deep convolutional neural network (CNN) utilized for classifying images. VGG-16 is characterized as having a simple structure, with smooth recognition and implementation. The VGG16’s parameters commonly include 16 layers, along with 13 convolutional layers and 3 fully connected layers, as shown in Figure 2. The layers were prepared into blocks, with every block composed of more than one convolutional layer observed through a max-pooling layer for downsampling. Table 1 shows the hyper-parameters for the VGG-16 model.

Choosing block3-conv3 in VGG-16 for feature extraction in medical imaging tasks such as brain tumor classification is based on a balance between local features and global context (tumor boundaries, structural patterns) for the following reasons [27]:Receptive field at Block 3 (24 × 24 pixels).The capture of tumor sub-regions.Fewer channels (256) vs. Block 4/5 (512), reducing memory overhead.Block 3 heatmaps precisely highlight tumor margins (for segmentation).

### 3.3. VIT

[Input MRI] → [Patch Embedding] → [Position Embedding] → [Transformer Encoder Blocks] → [MLP Head] → [Class Scores] (P×P patches) (Learnable pos codes) (L × [MSA + MLP])

Patch Embedding: Given an input image, we split it into *N* non-overlapping patches of size P×P:$$X_{p}\in R^{N\times (P^{2}\cdot C)},\;N=\frac{H\times W}{P^{2}}$$Each patch is linearly projected to dimension *D* using a learnable matrix E:$$z_{0}=\left[x_{p}^{1}E,x_{p}^{2}E,\ldots ,x_{p}^{N}E\right]+E_{\mathrm{pos}},\;E\in R^{(P^{2}\cdot C)\times D}$$
where $E_{\mathrm{pos}}$ denotes learnable positional embeddings.Multi-Head Self-Attention: For each head *h* in *H* parallel attention heads,$$\begin{array}{cc} Q_{h} & =zW_{h}^{Q},\;K_{h}=zW_{h}^{K},\;V_{h}=zW_{h}^{V}\\ \mathrm{Attention}(Q_{h},K_{h},V_{h}) & =\mathrm{softmax}\left(\frac{Q_{h}K_{h}^{T}}{\sqrt{D/H}}\right)V_{h} \end{array}$$The outputs are concatenated and projected:$$\mathrm{MSA}\left(z\right)=\mathrm{Concat}\left[{\mathrm{head}}_{1},\ldots ,{\mathrm{head}}_{H}\right]W^{O}$$
where $W^{O}\in R^{D\times D}$.Classification Head: The [CLS] token embedding $z_{0}^{0}\in R^{D}$ is used for classification:$$y=\mathrm{MLP}\left(z_{0}^{0}\right)=W_{2}\left(\mathrm{GELU}\left(W_{1}z_{0}^{0}+b_{1}\right)\right)+b_{2}$$
where $W_{1}\in R^{D\times 4D}$, and $W_{2}\in R^{4D\times K}$ (*K* classes).

### 3.4. FTVT-B16

The structure of the vision transformer models, such as ViT with a “B” version and sixteen layers, includes 12 transformer blocks [28]. The Vit/B16 models were pre-trained on the Image-Net datasets using a Adam optimizer and loss entropy feature within 10 epochs with an input image of 224 × 224 pixels, as shown in Table 2. The FTVT/B16 model consists of the ViT/B16 model with a custom classifier head that contains BN, ReLU, and dropout layers [29]. These layers enhance the capacity model to discover critical functions applicable to the direct class. The parameters are initialized randomly with the prevailing VIT/b16 model parameters [23]. The collection of introduced layers starts off with those evolved via BN, linear transformer layers, ReLU, and dropout layers. At last, the function vector is converted to the output area through linear layers, by figuring out the range of output lessons considering brain tumor types [30]. Figure 3 shows the FTVT architecture.

### 3.5. VGG16- FTVTB16

The typical structure of VGG-16 with fine-tuned vision transformer models (shown in Algorithm 1) with hyper-parameters is shown in Table 3:The input image is split into several patches.The images are flattened and assigned a labeling class.The outcomes of the transformer community are dispatched to the multilayer perception modules.Constructing the Fused Model: Constructing the model fusion of VGG-16 and FTVT-b16 includes many steps:Feature Extraction: Initial function extraction takes the area through the VGG-16 backbone, which analyzes the MRI images and extracts deep features.Attention Mechanism: The input function of the FTVT-b16 model makes use of its self-attention layers to create awareness for salient features and contextual cues that signify distinctive tumor types.Integration Layer: An integration layer merges the outputs of each model at the same time as the leveraging strategies, including concatenation or weighted averaging to create a unified feature representation.Classification: The final features from the mixing layer are fed into fully connected layers for the final types of tumors, including gliomas, meningiomas, and pituitary tumors.
**Algorithm 1** Hybrid VGG-16 and FTVT-B16 for Brain Tumor Classification.1:$X\in R^{H\times W\times 3}$: Input MRI volume2:$f_{\mathrm{VGG}}$: Pretrained VGG-16 up to block3_conv33:$f_{\mathrm{FTVT}}$: Fine-tuned Vision Transformer (ViT-B/16)4:*C*: Number of tumor classes      $y\in {\left[0,1\right]}^{C}$ Class probability vector **Feature Extraction**      *VGG-16 Pathway:*
$$F_{\mathrm{VGG}}=f_{\mathrm{VGG}}\left(X\right)\in R^{\frac{H}{8}\times \frac{W}{8}\times 256}$$*FTVT-B16 Pathway:*P=PatchEmbed(X,p=16)(Non-overlapping16×16patches)$$F_{\mathrm{FTVT}}=f_{\mathrm{FTVT}}\left(P\right)\in R^{N\times D}\;\left(N=\frac{HW}{256},D=768\right)$$$$F_{\mathrm{FTVT}}\leftarrow \mathrm{Reshape}\left(F_{\mathrm{FTVT}},\frac{H}{16},\frac{W}{16},D\right)$$$$F_{\mathrm{FTVT}}\leftarrow {\mathrm{Conv}}_{1\times 1}\left(F_{\mathrm{FTVT}}\right)\in R^{\frac{H}{8}\times \frac{W}{8}\times 256}$$**Cross-Modal Fusion:**$$G=\sigma (W_{g}\left[\mathrm{GAP}\left(F_{\mathrm{VGG}}\right);\mathrm{GAP}\left(F_{\mathrm{FTVT}}\right)\right]+b_{g})$$$$F_{\mathrm{fused}}=G⊙F_{\mathrm{VGG}}+\left(1-G\right)⊙F_{\mathrm{FTVT}}\in R^{\frac{H}{8}\times \frac{W}{8}\times 256}$$**Classification:**$$z={\mathrm{Global}}_{{\mathrm{Avg}}_{\mathrm{Pool}}}\left(F_{\mathrm{fused}}\right)\in R^{256}$$$$y=\mathrm{Softmax}(W_{c}z+b_{c})$$      **Loss Computation:**
$$L=\alpha L_{\mathrm{CE}}\left(y,y_{\mathrm{true}}\right)+\beta {∥W_{g}∥}_{2}$$

### 3.6. Dataset

The Kaggle database website (https://www.kaggle.com/datasets/masoudnickparvar/brain-tumor-mri-dataset, accessed on 19 September 2023) has the open access dataset used in this study [24]. Gliomas, meningiomas, pituitary tumors, and no tumor are four distinct groups into which the 7023 images of the human brain captured via MRI are classified. This dataset’s main goal is to help researchers create a highly accurate model for the detection and classification of brain tumors. A demographic representation of the Kaggle dataset is shown in Figure 4.

The second dataset, named Brain Tumour Detection 2020 (BR35H), is an MRI dataset obtained from the Kaggle website (kaggle.com/datasets/ahmedhamada0/brain-tumour-detection, accessed on 19 September 2023). We refer to this dataset as dataset no.2, consisting of two classes, with 1500 images of the tumor class and 1500 images of normal or non-tumor cases.

## 4. Experimental Results

The overall accuracy of the proposed framework may be computed using preferred metrics, which include the accuracy and precision, as listed in Table 4, Table 5 and Table 6. The experiments implied that the hybrid structure surpasses the overall performance of both standalone versions, specifically in dealing with many tumor shows and noise in MRI images.

The training and execution of the three models employed Google Co-laboratory (Co-lab), which is totally based on Jupyter Notebooks, a virtual GPU powered with the aid of an Nvidia Tesla K80 (manufactured by NVIDIA Corporation, Santa Clara, CA, USA) with 16 GB of RAM, the Keras library, and TensorFlow. Our proposed models, employed datasets, and all source codes are publicly available at https://github.com/mahmoudnasseraboali/VGG16-FTVTB16-for-brain-tumor-classifcation, accessed on 1 July 2025. Table 1 and Table 2 summarize the experimental results of the three models (VGG-16, FTVT-B16, and VGG-16-FTVT-B16). Figure 5 shows the accuracy according to each class of the dataset. Figure 6 shows the overall accuracy of the suggested models. Table 7 shows the computational complexity of three models (VGG-16, FTVT-B16, hybrid model). Table 8 explained Paired *t*-test results of hybrid model (five-fold CV) on dataset no.1.

### 4.1. Evaluation Metrics

Precision measures how often the model correctly predicted the disease, and it is calculated using the following equation:$$Precision=\frac{TP}{TP+FP}$$

Recall is defined as the number of true positives (TP) divided by the number of true positives plus the number of false negatives (FN) using the following equation:$$Recall=\frac{TP}{TP+FN}$$F1-score: A weighted average of true positive rates (recall rates) using the following formula:$$F1-score=\frac{2*precision*recall}{precision+recall}$$Focal loss: is a dynamically weighted variant of the standard Cross-Entropy Loss, designed to address class imbalance by down-weighting easy-to-classify examples and focusing training on difficult misclassified samples. The Focal Loss for class-imbalanced classification is defined as(1)$$L_{\mathrm{FL}}=-\sum_{c=1}^{C}\alpha_{c}\cdot {\left(1-p_{c}\right)}^{\gamma }\cdot y_{c}\cdot \log\left(p_{c}\right)$$
where

$\alpha_{c}$ balances class importance;${\left(1-p_{c}\right)}^{\gamma }$ down-weights easy examples;γ controls the focusing strength.

A confusion matrix was normalized by dividing every element’s cost in each class, improving the visible illustration of misclassifications in every class, which shows that G, P, M, and no-tumor, or 0, 1, 2, and 3, confer with glioma, pituitary tumor, meningioma, and no tumor, respectively. Distinct values may be used from the confusion matrix to illustrate the overall classification accuracy and recall for every class in the dataset. Figure 7 and Figure 8 illustrate the confusion matrix for each model.

### 4.2. Visualization Analysis Using Grad-CAM

An effective technique for illustrating which parts of an input image are crucial is used in the brain tumor classification with the proposed hybrid model of VGG-16 and FTVT-B16. Predictions were carried out using GRAD-CAM (Gradient-weighted Class Activation Mapping). GRAD-CAM aids in highlighting the discriminative regions the model focuses on when employed for brain tumor classification, as shown in Figure 9 on dataset no.1.

The classifications of brain tumors using GRAD-CAM analysis are as follows:No tumors: no localized highlights, tumor diffuse, or potential misleading highlights in artifacts or normal variants.Pituitary tumors: The heatmap focuses on the area around the pituitary gland; if it is large, it may indicate compression of the optic chiasm.Glioma tumors: The heatmap emphasizes edema and the tumor core.Meningioma tumors: The hheatmap emphasizes strong activation close to the dural tail (if present) and smooth, rounded tumor borders.

## 5. Results and Discussion

This study concerned classifying brain tumor types through a preprocessing stage using a hybrid approach of VGG-16 with fine-tuning vision transformer models, and by comparing overall performance metrics such as the accuracy, precision, and recall. The fusion model demonstrates 99.46% for classification accuracy, surpassing the performance of standalone VGG-16 (97.08%) and FTVT-b16 (98.84%). In this paper, we introduce a hybrid of VGG-16 with fine-tuned vision transformer models to improve brain tumor classification using MR images. The suggested framework demonstrated outstanding performance, suggesting a promising direction of automated brain tumor detection in terms of sensitivity, precision, and recall. Figure 10 shows the performance metrics (precision, recall, F1-score, specificity) from the fusion of the VGG-16 and FTVT-b16 models. Figure 11 shows the accuracy and loss over epochs for the proposed hybrid model.

Table 9 Comparison of related work with achieved accuracy on the same dataset no.1 (Kaggle dataset). Table 10 Comparison of the obtained accuracy with that achieved in past work on the same dataset (dataset no.2 (BR35H)).

The hybrid model performs better than current benchmarks when compared to modern approaches. For example, it outperformed the parallel deep convolutional neural network (PDCNN, 98.12%; Rahman et al., 2023 [17]) and the Inception-QVR framework (99.2%; Amin et al., 2022 [22]), both of which depend on homogenous designs, in terms of accuracy. While pure ViTs require large amounts of training data (Khan et al., 2022 [13]) and conventional CNNs like VGG-19 are limited by their localized receptive fields (Mascarenhas & Agarwal, 2021 [11]), the suggested hybrid makes use of FTVT-b16’s ability to resolve long-range pixel interactions and VGG-16 effectiveness in hierarchical feature extraction. Confusion matrix analysis also showed that 15 pituitary cases were incorrectly classified by the solo VGG-16, most likely as a result of its incapacity to contextualize anatomical relationships outside of local regions. To illustrate the transforming function of transformer-driven attention in disambiguating heterogeneous tumor appearances, the hybrid model, on the other hand, decreased misclassifications to two cases per class (Tummala et al., 2022 [23]).

The model’s balanced class-wise measures and excellent specificity (99.82%) offer significant promise for improving diagnostic workflows from a clinical standpoint. The methodology tackles a widespread issue in neuro-oncology, where incorrect diagnosis can result in unnecessary surgical procedures, by reducing false positives (van den Bent et al., 2023 [1]). For instance, the model’s incorporation of both global and local variables improves the accuracy of distinguishing pituitary adenomas from non-neoplastic cysts, a task that has historically been subjective. Additionally, the FTVT-b16 component produces attention maps that show the locations of tumors, providing radiologists with visually understandable explanations.

## 6. Conclusions

Brain tumor classification on MRI is considered a critical issue in neuro-oncology, necessitating advanced computational frameworks to enhance diagnostic performance. This paper proposed a hybrid deep learning framework that extracts features using VGG-16 with the global contextual modeling of FTVT-b16, a fine-tuned vision transformer (ViT), to address the limitations of standalone convolutional neural networks (CNNs) and transformer-based architectures. The proposed framework was tested on a dataset of 7023 MRI images spanning four classes, glioma, meningioma, pituitary, and no-tumor. It achieved a classification accuracy of 99.46% on Kaggle dataset no.1 and achieved 99.90% on Br35H dataset no.2, surpassing both VGG-16 (97.08%) and FTVT-b16 (98.84%). Key performance metrics, including precision (99.43%), recall (99.46%), and specificity (99.82%), underscore the model’s robustness, particularly in reducing misclassifications of anatomically complex cases such as pituitary tumors. Table 11 shows ROC-AUC comparison (macro-averaged) on dataset no1.

The “black-box” constraints of traditional deep learning systems were addressed by combining CNN-derived local features with ViT-driven global attention maps, which improved interpretability through displayed interest regions and increased diagnostic precision. This dual feature gives radiologists both excellent accuracy and useful insights into model decisions, which is in line with clinical processes.

Despite these developments, generalization is limited by the study’s reliance on a single-source dataset because demographic variety and scanner variability were not taken into consideration. Multi-institutional validation should be given top priority in future studies in order to evaluate robustness across diverse demographics and imaging procedures. Furthermore, investigating advanced explainability tools (e.g., Grad-CAM, SHAP) and multi-modal data integration (e.g., diffusion-weighted imaging, PET) could improve diagnostic value and clinical acceptance. To advance AI-driven precision in neuro-oncology, real-world application through collaborations with healthcare organizations and benchmarking against advanced technology like transformers will be essential.

In conclusion, this paper lays a foundational paradigm for hybrid deep learning in medical image analysis, by showing that the fusion of a CNN and ViT can bridge the gap between computational innovation and clinical utility. By enhancing both accuracy and interpret ability, the proposed framework has great potential to improve patient outcomes through earlier, more accurate diagnosis by increasing both accuracy and interpretability. 

## Figures and Tables

**Figure 1 diagnostics-15-02014-f001:**
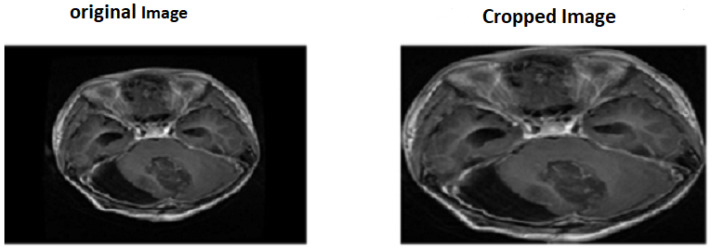
Applying median filter and cropped image.

**Figure 2 diagnostics-15-02014-f002:**
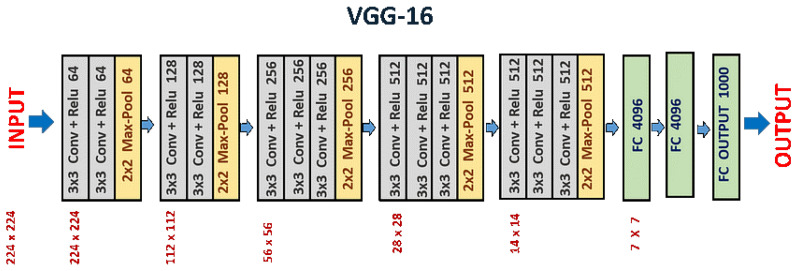
VGG-16 architecture.

**Figure 3 diagnostics-15-02014-f003:**
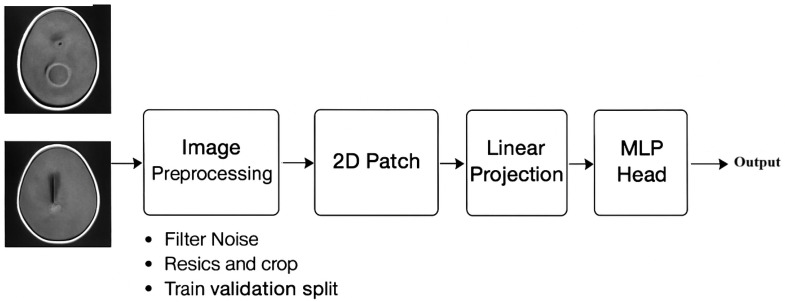
FTVT architecture.

**Figure 4 diagnostics-15-02014-f004:**
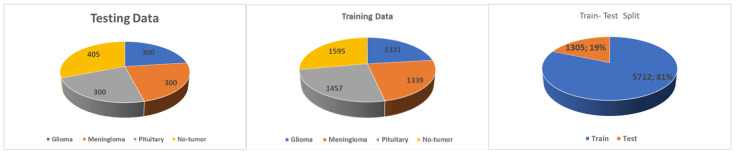
Performance metrics of three three proposed models.

**Figure 5 diagnostics-15-02014-f005:**
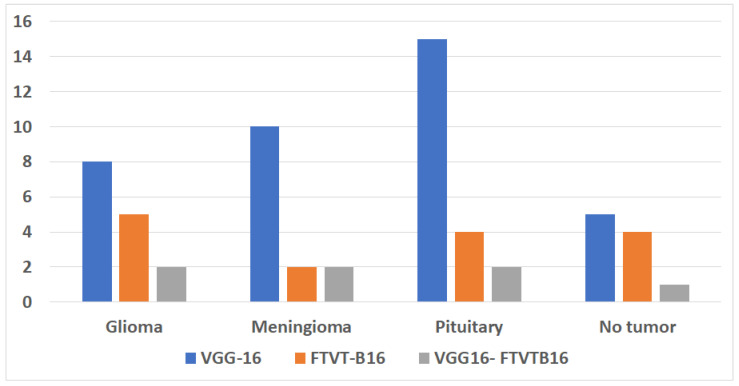
Accuracy of four class tumor types on each model.

**Figure 6 diagnostics-15-02014-f006:**
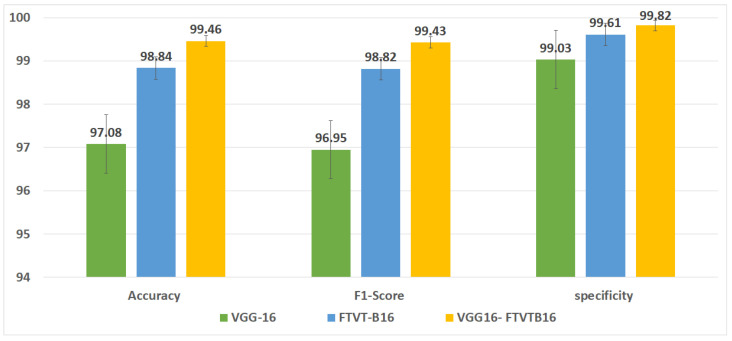
Performance metrics of three proposed models.

**Figure 7 diagnostics-15-02014-f007:**
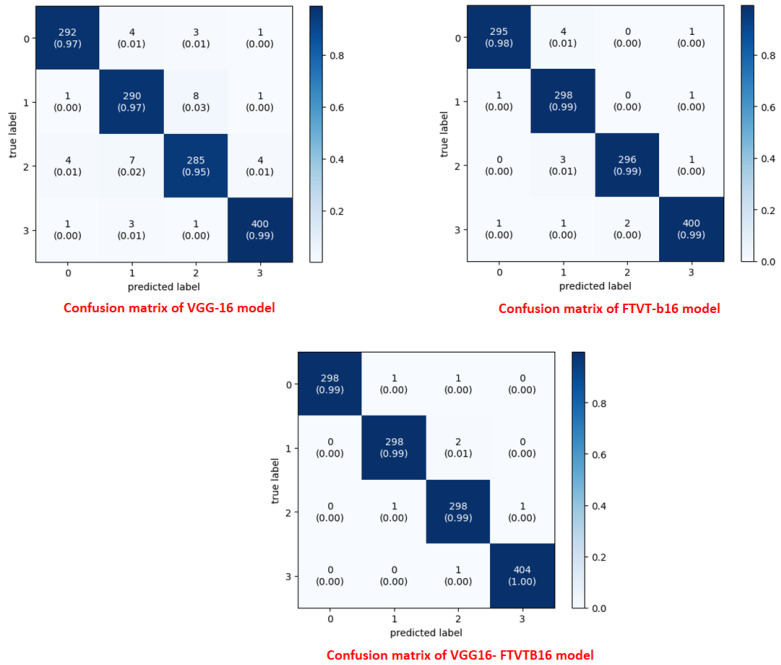
Confusion matrix for the three proposed models on dataset no.1.

**Figure 8 diagnostics-15-02014-f008:**
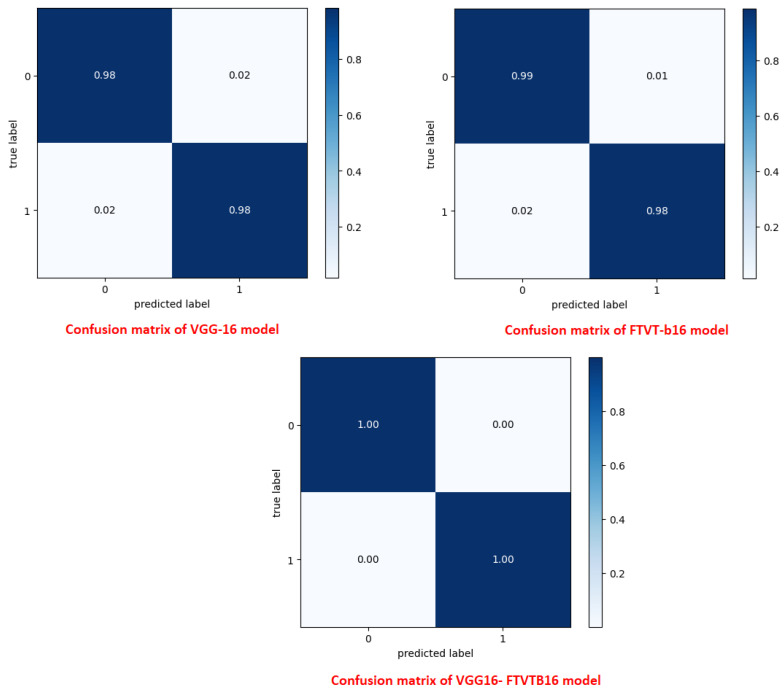
Confusion matrix for the three proposed models on dataset no.2.

**Figure 9 diagnostics-15-02014-f009:**
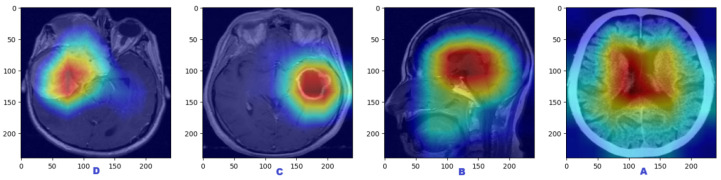
Visualization analysis using grad-CAM on dataset no.1. (**A**: no tumor; **B**: pituitary tumors; **C**: glioma tumors; **D**: meningioma tumors).

**Figure 10 diagnostics-15-02014-f010:**
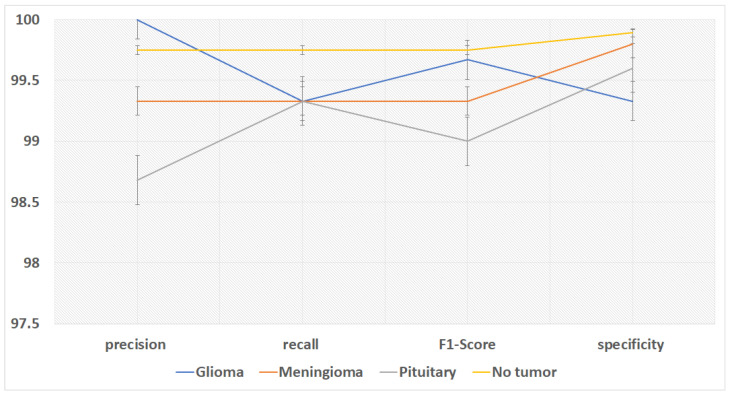
Overall performance of proposed hybrid model according to tumor types.

**Figure 11 diagnostics-15-02014-f011:**
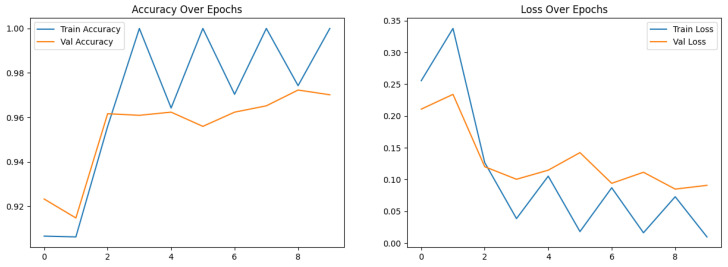
Accuracy and loss over epochs for the proposed hybrid model.

**Table 1 diagnostics-15-02014-t001:** Hyper-parameters for VGG-16 model.

Parameter	Value
Input Size	224 × 224 × 3
Activation Function	ReLU
Dropout	0.5
Learning rate	1 × $10^{-4}$
Batch Size	16
Epochs	10

**Table 2 diagnostics-15-02014-t002:** Hyper-parameters for FTVT-B16 model.

Parameter	Value
Input Image Size	224 × 224 × 3
Patch Size	16 × 16
Optimizer	Adam
Learning Rate	1 × $10^{-4}$
Patch Embedding Dim	768
Positional Embeddings	Learnable
Number of Layers	12
Attention Heads	12
MLP Hidden Dim	3072 (4 × 768)
Dropout Rate	0.1
Layer Normalization	linear transformer
Epochs	10

**Table 3 diagnostics-15-02014-t003:** Hyper-parameters for proposed hybrid model.

Parameter	Value	Rationale
VGG-16 freeze	Up to Block 3	Preserves low/mid-level features
FTVT patch size	16 × 16	Standard ViT configuration
Fusion dim	256	Bottleneck for efficiency
Learning rate	$3\times 10^{-5}$	AdamW optimizer
Loss weights (α,β)	0.9, 0.1	Balance classification regularization

**Table 4 diagnostics-15-02014-t004:** Experimental results on dataset no.1.

Model	Class	Accuracy	Precision	Recall	F1-Score	Specificity
MobileNetV2	Glioma	95.00	95.64	94.37	94.86	94.49
	Meningioma	96.00	94.56	94.43	94.09	94.55
	Pituitary	96.33	94.92	94.03	94.84	95.01
	No tumor	97.78	94.32	94.37	94.62	94.23
ResNet50	Glioma	96.33	96.40	95.97	95.74	96.02
	Meningioma	96.00	96.54	96.50	95.99	96.51
	Pituitary	96.66	96.43	96.74	96.06	96.56
	No tumor	98.02	97.72	98.02	98.02	97.93
DenseNet	Glioma	95.30	95.33	95.33	95.34	95.90
	Meningioma	97.00	96.84	96.80	96.20	96.50
	Pituitary	96.66	96.23	96.34	96.16	96.44
	No tumor	98.50	98.50	98.52	98.50	98.50
EfficientNet	Glioma	96.30	96.23	96.31	96.33	96.50
	Meningioma	97.20	96.84	96.84	96.84	96.90
	Pituitary	97.50	97.50	97.44	96.94	96.94
	No tumor	98.60	98.60	98.00	98.00	98.00
VGG-16	Glioma	97.33	97.99	97.33	97.66	99.40
	Meningioma	96.67	95.39	96.67	96.03	98.61
	Pituitary	95.01	95.96	95.00	95.48	98.81
	No tumor	98.75	98.52	98.77	98.64	99.33
FTVT-B16	Glioma	99.10	99.33	98.33	98.83	99.80
	Meningioma	99.12	97.39	99.33	98.35	99.20
	Pituitary	98.65	99.33	98.67	99.00	99.80
	No tumor	98.66	99.26	99.01	99.13	99.67
VGG16	Glioma	99.30	100.0	99.33	99.67	99.33
FTVTB16	Meningioma	99.30	99.33	99.33	99.33	99.80
	Pituitary	99.34	98.68	99.33	99.00	99.60
	No tumor	99.72	99.75	99.75	99.75	99.89

**Table 5 diagnostics-15-02014-t005:** Experimental results on dataset no.2.

Model	Class	Accuracy	Precision	Recall	F1-Score	Specificity
MobileNetV2	Tumor	97.09	97.09	97.34	97.21	97.21
	No tumor	97.34	97.34	97.09	97.21	97.21
ResNet50	Tumor	97.34	97.28	97.40	97.34	97.34
	No tumor	97.00	97.40	97.28	97.34	97.34
DenseNet	Tumor	97.90	97.78	97.85	97.82	97.82
	No tumor	97.90	97.85	97.78	97.82	97.82
EfficientNet	Tumor	98.04	98.04	97.89	98.01	98.01
	No tumor	97.98	97.98	98.04	97.89	97.89
VGG-16	Tumor	98.19	98.23	98.17	98.23	98.23
	No tumor	98.17	98.17	98.23	98.23	98.23
FTVT-B16	Tumor	98.43	98.43	98.55	98.49	98.49
	No tumor	98.55	98.55	98.43	98.49	98.49
VGG16-FTVTB16	Tumor	99.88	99.88	99.88	99.88	99.88
	No tumor	99.90	99.90	99.90	99.90	99.90

**Table 6 diagnostics-15-02014-t006:** Experimental results of proposed model.

Dataset	Model	Accuracy	Precision	Recall	F1-Score	Specificity	Mean
Dataset no.1	MobileNetV2	96.70	96.50	96.50	96.20	96.00	96.70
	ResNet50	96.99	96.90	99.90	96.90	96.90	96.90
	DenseNet	97.20	97.20	97.20	97.20	97.20	97.20
	EfficientNet	97.50	97.50	97.20	97.20	97.20	97.20
	VGG16	97.80	97.08	96.95	99.03	97.45	97.32
	FTVT-B16	98.10	98.84	98.82	99.61	98.65	98.52
	VGG16-FTVTB16	99.46	99.46	99.43	99.82	99.46	99.30
Dataset no.2	MobileNetV2	97.21	97.21	97.21	97.21	97.21	97.21
	ResNet50	97.34	97.34	97.34	97.34	97.34	97.34
	DenseNet	97.81	97.81	97.81	97.81	97.81	97.81
	EfficientNet	98.07	98.07	98.07	98.07	98.07	98.07
	VGG16	98.19	98.19	98.19	98.19	98.19	98.19
	FTVT-B16	98.48	98.48	98.48	98.48	98.48	98.48
	VGG16-FTVTB16	99.90	99.90	99.90	99.90	99.90	99.90

**Table 7 diagnostics-15-02014-t007:** Computational complexity of the models.

Model	FLOPs	Parameters	GPU Mem (BS = 16)
MobileNetV2	0.3	3.4	1.5 GB
ResNet50	13.8	25.6	2.1 GB
DenseNet	2.9	8	1.8 GB
EfficientNet	0.39	5.3	1.2 GB
VGG-16	15.5	138 M	2.5 GB
ViT-B16	17.6	86 M	3.0 GB
Hybrid (Proposed)	33–38	225–230 M	6.0 GB

**Table 8 diagnostics-15-02014-t008:** Paired *t*-test results of hybrid model (five-fold CV) on dataset no.1.

Metric	Value
t-Statistic	3.75
*p*-value	0.0001
Mean Improvement	0.014

**Table 9 diagnostics-15-02014-t009:** Comparison of related work with achieved accuracy on the same dataset no.1 (Kaggle dataset).

Author	Year	Method	Accuracy	Epoches
Saleh et al. [15]	2020	Xception, ResNet50, InceptionV3,VGG16, and MobileNet	maximum: 98.75%	20
Kokila et al. [14]	2021	CNN	97.87%	20
Ullah et al. [21]	2022	CNN for feature extraction and SVM for classification	98.91%	10
Amin et al. [22]	2022	inceptionv3, quantum variational classifier (QVR)	99.2%	50
Sharma et al. [20]	2023	VGG19	98.00%	20
Ozkaraca et al. [16]	2023	d VGG16Net and DenseNet	97.00%	10
Rahman et al. [17]	2023	parallel deep convolutional neural network (PDCNN)	98.12%	50
Alyami et al. [19]	2024	AlexNet and VGG1 with SVM	99.1%	10
Reyes et al. [31]	2024	combined VGG, ResNet	is 97.9%	20
Sachdeva et al. [32]	2024	ResNet50, InceptionV3, Xception, DenseNet	maximum accuracy: 97.32%	10
Proposed model	2025	VGG-16	97.08%	10
Proposed model	2025	FTVT-B16	98.84%	10
Proposed model	2025	VGG16-FTVTB16	99.46%	10

**Table 10 diagnostics-15-02014-t010:** Comparison of related work with achieved accuracy on the same dataset no.2 (BR35H).

Author	Year	Method	Accuracy
Hamada et al. [33]	2020	CNN models	97.5%
Asmaa et al. [34]	2021	CNN with augmented images	98.8%
Amran et al. [35]	2022	AlexNet, MobileNetV2	99.51%
Falak et al. [36]	2023	Keras Sequential Model (KSM)	97.99%
eman et al. [18]	2024	Combination of CNN with EfficientNetV2B3 and KNN classifier	99.83%
Proposed model	2025	VGG16-FTVTB16	99.90%

**Table 11 diagnostics-15-02014-t011:** ROC-AUC comparison (macro-averaged) on dataset no.1.

Model	Glioma	Meningioma	Pituitary	Normal	Macro AUC
VGG16	0.973	0.966	0.950	0.987	0.969
FTVTB16	0.991	0.991	0.986	0.986	0.988
VGG16-FTVTB16	0.993	0.993	0.993	0.997	0.994

## Data Availability

The data collected and analyzed during this study are available from the corresponding author upon reasonable request.

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
