# Peer review of "Hybrid of VGG-16 and FTVT-b16 Models to Enhance Brain Tumors Classification Using MRI Images"

_diagnostics, 2025, doi:10.3390/diagnostics15162014_

Round 1

Reviewer 1 Report

Comments and Suggestions for Authors

The manuscript has the following concerns.

(1) The abstract should be refined, and key highlights of the method should be included.

 (2) After the Introduction section contribution of the method should be added in points.

(3) The literature review section must include some recent methods and add a paragraph on the weaknesses of the previous methods.

(4) Figure 2 should be revised, and the connection between each layer should be provided.

(5) A mathematical description of the method must be added, and each variable should be defined.

(6) It will be better to add an algorithm after the methodology section.

(7) Figure of the ViT model must be included, and it should be described how patch embedding and attention are calculated.

(8) Details of the loss function should be elaborated.

(9) How many layers of the transformer did you utilized in the study, and what was the feature dimension?

(10) Training and validation loss curves can be added to check for the overfitting problem.

(11) In the discussion section, you must compare the proposed model results with recent methods, ResNeXtV2, Inception v4, DaViT, and Swin Transformer.

(12) After that, add an ROC-based comparison of the stated methods.

(13) Add a statistical analysis using a paired t-test to validate the performance.

Author Response

Please Check the attached response

Reviewer 2 Report

Comments and Suggestions for Authors

1. In the final paragraphs of the Introduction section, the contributions of the study should be listed in bullet points.
2. Why was VGG16 chosen as the backbone network? Explicitly state the advantages of VGG16 compared to models like DenseNet, EfficientNet, MobileNet, ResNet, etc.
3. When evaluated as an innovative method, the proposed approach appears to be weak.
4. Please correct the class names in Figure 3. They should be: "no_tumor, gliomas, meningiomas, and pituitary tumors." Also, the spelling of "output" is incorrect. Please correct these as well.
5. A new dataset is required. An additional dataset should be incorporated into the study, and experimental analyses should be conducted.
6. Why was the dataset split as 80-20? Would it not be better to use the original train-test split of the dataset?
7. What are the hyperparameters?
8. Under Section 4, separate subheadings titled "Hyperparameter Settings" and "Evaluation Metrics" should be added. Then, under these subheadings, the hyperparameters and evaluation metrics should be explained. Providing the formulas for the evaluation metrics would also be beneficial.
9. How many epochs were used to train your experimental models?
10. Class-based accuracy values should be added to Table 1. Additionally, recall and precision values should be included in Table 2.
11. What is the computational complexity and memory size of the proposed model?
12. In your experimental studies, results should also be obtained using models like ResNet, DenseNet, EfficientNet, MobileNet instead of VGG16. These results should be presented in a table.
13. Curves of accuracy-loss should be included.
14. Table 2 appears to be an ablation analysis. Grad-CAM analysis should be performed for each model.
15. Does Table 3 use the same dataset for all studies? To ensure a fair comparison, the hyperparameters, such as the number of epochs for each model, should be added to the table.

Author Response

Please find the attachment response letter

Reviewer 3 Report

Comments and Suggestions for Authors

In this work, the authors combined VGG-16 and a fine-tuned Vision Transformer (FTVT-b16) to improve the classification of brain tumors in MRI images. The framework was trained on a public Kaggle dataset of 7,023 MRI images across four tumor classes (glioma, meningioma, pituitary, and no-tumor). It achieved a state-of-the-art accuracy of 99.46%, outperforming individual VGG-16 (97.08%) and FTVT-b16 (98.84%) models. The paper also emphasizes the model's interpretability through attention maps and its clinical relevance. Future work suggests multi-institutional validation and computational optimization for broader deployment.

-    The authors are suggested to proofread the paper for grammatical errors, awkward phrasing, and inconsistent terminology (e.g., "fine structures are degraded", "the first-class category performance", "images were flattened and furnished with class labels"). 
-    The Introduction and Discussion sections repeat similar information (e.g., clinical importance, tumor characteristics). 
-    The flow of Sections 3.2 to 3.3.1 is disorganized. The descriptions of VGG-16 and FTVT-B16 should clearly separate architecture, pretraining, and implementation details.
-    The authors are suggested to provide clearer justification for selecting specific layers (e.g., why features from layer 3 of VGG-16 are chosen).
-    It is recommended to add/discuss the following work in introduction section: 1:Optimal feature extraction and ulcer classification from WCE image data using deep learning 2: Medical imaging fusion techniques: a survey benchmark analysis, open challenges and recommendations 3: A deep learning and handcrafted based computationally intelligent technique for effective COVID-19 detection from X-ray/CT-scan imaging
-     It is suggested to add the clarification related to the fusion methodology—whether feature-level or decision-level fusion was performed. The term “integration layer” should be better defined with architectural details or a schematic.
-    The training methodology lacks important details like data augmentation techniques, validation strategy (e.g., cross-validation or hold-out), epoch count, and learning rate schedules beyond a vague mention of Adam optimizer.
-    The authors are suggested to add the details about training time and batch sizes.
-    The paper does not analyze class imbalance or demographic representation within the dataset. 
-    Some figures (e.g., Fig. 1, Fig. 3) are referred to without descriptive captions or meaningful discussion in the text.

Author Response

(The authors gave the same response as above.)

Round 2

Reviewer 1 Report

Comments and Suggestions for Authors

The authors fail to incorporate all the concerns. Still novelty is very less

Author Response

Please find the attached response Letter

Reviewer 2 Report

Comments and Suggestions for Authors

1. Please change "Resent50" to "ResNet50" in Table 6.
2. Add the results of models such as ResNet50, DenseNet, EfficientNet, MobileNet, and MobileNetV2 to Tables 4, 5, 6, and 7.
3. The epoch values were added as a column in Table 8. However, the epoch values for the models are not written.
4. Visual results of the GRAD-CAM analysis should be included in the paper. Please add visual results of GRAD-CAM analysis for each tumor class.
5. Accuracy values should be added to Tables 4, 5, and 6.
6. A fair comparison between the two datasets and the studies in the literature should be made in Table 8.
7. Include confusion matrices for both datasets.
8. Is the accuracy value in Table 8 the macro accuracy or the overall accuracy?

Author Response

(The authors gave the same response as above.)

Reviewer 3 Report

Comments and Suggestions for Authors

The authors failed to comply with all the suggested improvements. It is required to look into all the comments and address all the suggestions: For example, comment # 5. 

Author Response

(The authors gave the same response as above.)

Round 3

Reviewer 1 Report

Comments and Suggestions for Authors

The manuscript can be accepted.

Author Response

Thanks for your Valuable feedback

Reviewer 2 Report

Comments and Suggestions for Authors

I thank the authors for the changes they have made.

Author Response

Thanks for your Valuable feedback